# CNN Architectures and Feature Extraction Methods for EEG Imaginary Speech Recognition

**DOI:** 10.3390/s22134679

**Published:** 2022-06-21

**Authors:** Ana-Luiza Rusnac, Ovidiu Grigore

**Affiliations:** Department of Applied Electronics and Information Engineering, Faculty of Electronics, Telecommunications and Information Technology, Polytechnic University of Bucharest, 060042 Bucharest, Romania

**Keywords:** imaginary speech, convolutional neural network, electroencephalography, signal processing, Kara One database

## Abstract

Speech is a complex mechanism allowing us to communicate our needs, desires and thoughts. In some cases of neural dysfunctions, this ability is highly affected, which makes everyday life activities that require communication a challenge. This paper studies different parameters of an intelligent imaginary speech recognition system to obtain the best performance according to the developed method that can be applied to a low-cost system with limited resources. In developing the system, we used signals from the Kara One database containing recordings acquired for seven phonemes and four words. We used in the feature extraction stage a method based on covariance in the frequency domain that performed better compared to the other time-domain methods. Further, we observed the system performance when using different window lengths for the input signal (0.25 s, 0.5 s and 1 s) to highlight the importance of the short-term analysis of the signals for imaginary speech. The final goal being the development of a low-cost system, we studied several architectures of convolutional neural networks (CNN) and showed that a more complex architecture does not necessarily lead to better results. Our study was conducted on eight different subjects, and it is meant to be a subject’s shared system. The best performance reported in this paper is up to 37% accuracy for all 11 different phonemes and words when using cross-covariance computed over the signal spectrum of a 0.25 s window and a CNN containing two convolutional layers with 64 and 128 filters connected to a dense layer with 64 neurons. The final system qualifies as a low-cost system using limited resources for decision-making and having a running time of 1.8 ms tested on an AMD Ryzen 7 4800HS CPU.

## 1. Introduction

Communication is the basis of interpersonal relationships and is one of the most important ways to connect with other people and to express your needs and feelings. The most common forms of communication are writing or speaking, but the latter is the most natural mechanism involved in the transmission of thoughts. This relatively easy to gain ability is often taken for granted; however, it hides a complex mechanism. Speaking involves translating thoughts into the desired words and transmitting them with the help of motor neurons to a large number of muscles and joint components of the vocal tract that must be positioned differently for each spoken sound. This is why speech takes a large part of cortical motor homunculus [1].

Unfortunately, there are cases when this ability is lost, or the speech cannot be articulated due to some affections such as cerebral stroke, lock-down syndrome, amyotrophic lateral sclerosis, cerebral palsy, etc. In order to overcome this dysfunction, a series of alternative methods were proposed. The purpose of the research in this field was to find an easy and natural way of communication.

The activity of the brain can be measured using different methods such as electroencephalography (EEG), magnetoencephalography (MEG), electrocorticography (ECoG), functional magnetic resonance imaging (fMRI) and stereoelectroencephalography (sEEG). However, when it comes to developing brain computer interface systems (BCI), the most common methods for brain activity recording are EEG and MEG due to their considerable advantages of being non-invasive techniques and more accessible for signal acquisition. The ECoG signals are also widely used in BCI systems, even though they are invasive. The major advantage of the ECoG signals is the quality of the brain activity measurements by recording the signals directly from the cortex, eliminating in this way the attenuation given by the tissues between the cortex and the electrodes in comparison to EEG. fMRI signals are harder to acquire and more expensive than EEG and MEG, even though it is also a non-invasive method. Nevertheless, the best quality of brain activity measurements is collected using the sEEG technique because the electrodes are implanted deep into the brain. This method is the least used method for BCI due to the invasive approach.

In our study, we chose to focus on the EEG signals for their advantages in developing a low-cost, non-invasive portable device.

## 2. State of the Art

One of the first studies that tried to reconstruct the speech from EEG signals dates back to 1967 when the scientist Edmond M. Dewan [2] discovered that we can voluntarily control the alpha wave of the EEG signal. Starting from this point, the scientist used morse code in his developed system in order to obtain letters and, finally, conduct words.

Later studies also focused on creating words from letters for subjects to silently communicate with the computer. For example, in 2000, P. R. Kennedy et al. [3] used implanted neurotrophic electrodes on patients with amyotrophic lateral sclerosis (ALS) or brain stroke and obtained a functional system that uses the movement of a cursor as a form of communication. One of the system paradigms was to form words from letters by moving a cursor on the monitor and choosing the desired letter. Another similar approach was presented in [4] that concentrated on finding the trigger of P300 event-related potential (ERT) when the desired line and column of a matrix with letters and numbers were highlighted. Both methods work properly; however, these approaches represent an inconvenient way to communicate since it takes a long time to form a word.

Recent studies focused on finding patterns in EEG signals acquired during imaginary speaking of words or phonemes rather than finding a trigger, trying to obtain a more cursive way of communicating the thoughts. One attempt at unspoken speech recognition was made by Marek Wester in 2006 [5] for his PhD thesis, with results that reached 50% accuracy in multiple class classification. However, the group later revealed in [6] that the experiment process favored the results because the signal acquisition protocol assumed to speak or think the exact stimulus multiple consecutive times, and this accidentally creates temporal correlation in EEG signals. This was an important discovery in data acquisition protocol for further created databases.

In 2015, an open-source database acquired by Schunan Zhao and Frank Rudzicz at the Toronto Rehabilitation Institute was released [7]. The database contains signals collected from 14 healthy subjects during thinking and speaking of seven phonemes: /iy/, /uw/, /piy/, /diy/, /tiy/, /m/, /n/ and four words: “pat”, “pot”, “knew”, “gnaw”. This stimulus was chosen to have a relatively even number of vowels, plosives, and nasals as well as voiced and unvoiced phonemes. The researchers further created five binary classification tasks: consonant versus vocals (C/V), presence or absence of nasal (±Nasal), presence or absence of bilabial (±Bilabial), presence or absence of /iy/ phoneme (±/iy/) and presence or absence of /uw/ phoneme (±/uw/). In the conducted study, the researchers computed various statistical features over 10% of the segment windows with 50% overlap, including mean, median, standard deviation, variance, etc. (the details are specified in Table 1). They used the SVM-quad classifier and obtained maximum accuracy over the /uw/ phoneme: 79.16% and the minimum accuracy when classifying consonants versus vocals: 18.08%.

Later, in 2017, using the same database, the researchers Pengfei Sun and Jun Qin [8] conducted an experimental evaluation of three neural networks based on EEG-speech (NES) with the purpose of recognizing all the eleven phonemes. The three neural network models were: imagined EEG-speech (NES-I), biased imagined-spoken EEG-speech (NES-B) and gated imagined-speech (NES-G), with the last two introducing the EEG signals acquired during actual speech. The best results in this multi-classification problem were obtained using the NES-G network with an overall accuracy of 41.5%.

Another approach for the Kara One database binary task classification was proposed by Pramit Saha and Sidney Fels at the University of British Columbia [9]. In the developed study, the researchers used a mixed deep neural network strategy composed of a convolutional neural network (CNN), a long-short term network (LSTM) and a deep autoencoder. The hierarchical deep neural network used the cross-covariance matrix as the input feature matrix, with this method of feature extraction aiming to encode the connectivity of the electrodes. The obtained results increased the overall accuracy of the above binary tasks by 22.5%, achieving an average accuracy of 77.9% across the five known tasks [7].

However, when it comes to multi-classification of the phonemes and words, the results decrease significantly. In 2018 [10], a group of researchers introduced methods of speech recognition in their imaginary speech recognition from EEG signals using mel-cepstral coefficients (MFCC) as feature extraction and SVM classifier for recognition and broke the ice with an average accuracy of 20.80%—this value rising by 9% over the chance level. The results slightly improved when using MFCC for feature extraction and CNN as a classifier in the study [11]. The CNN neural network improved the overall accuracy, obtaining 24.19%.

Nevertheless, the highest accuracy over the multi-class classification of the Kara One phonemes and words was also obtained by the researchers from the University of British Columbia [12]. In their study, the researchers used the cross-covariance matrix (CCV) as feature extraction and a hierarchical combination of deep neural networks. In the first level of the final architecture of the classifier, a CNN was used to extract the spatial features from the covariance matrix. In parallel with CNN, they applied a temporal CNN (TCNN) to explore the hidden temporal features of the electrodes. Further, the latest fully connected layers from the CNN and TCNN were concatenated to compose a single feature vector, which was introduced to the second level of hierarchy consisting of a deep autoencoder (DAE). In the third level of hierarchy, they introduced the latent vector of DAE into an extreme gradient boost classification layer. The final neural network was first used to train the network for all six phonological tasks of Kara One and then to combine the gained information to further predict individual phonemes from all eleven categories.

Recent studies reported more encouraging results on the multi-class classification system of imaginary speech recognition. Developing an impressive database of eight different Russian words acquired from 270 subjects, the researchers [13] obtained a maximum accuracy of 85% when classifying the nine collected words and 88% for binary classification. The results were obtained using the frequency-domain of the signals and were classified with ResNet18 + 2GRU (gated recurrent unit).

Significant results for the imaginary speech recognition community were also obtained by using MEG signals. In 2020, Debadatta Dash, Paul Ferrari and Jun Wang [14] conducted a study based on MEG signals in order to recognize imagined and articulated speech of three different phrases of the English language. To achieve the final goal, the researchers used the discrete wavelet transform (DWT) in the feature extraction stage using a Daubechies (db)-4 wavelet with a seven-level decomposition. Further, they compared artificial neural networks (ANNs) and different configurations of CNNs. The best results were recorded using Spatial Spectral Temporal CNN, reaching an accuracy for the specific three classes of imaginary speech of 93.24%.

ECoG signals were also used for speech recognition and synthesis by Christian Herff et al. in [15]. The researchers managed to synthesize the vocal signals after analyzing motor, premotor and inferior frontal cortices and obtained an accuracy of 66.1% ± 6% in the correct identification of the word of 55 volunteered subjects. This approach offered very encouraging results for a real-time system; however, the brain signals were acquired in articulated speech (not imaginary speech), and the signals were collected using an invasive method.

Another recent study published in 2022 on ECoG signals for imaginary speech recognition was conducted by Thimotheé Proix et al. [16]. For binary classification using an SVM classifier, they managed to obtain an accuracy of over 60% for a patient-specific system. In the feature extraction stage, they used the analytic Morlet wavelet transform. The bands of interest were theta (4–8 Hz), low-beta (12–18 Hz), low-gamma (25–35 Hz) and broadband high-frequency activity (80–150 Hz).

An important role in the research community for EEG signal classification was also taken by the long-short term memory (LSTM) neural networks. LSTM neural networks are considered an improvement of the recurrent neural networks (RNN) due to the inclusion of the “gates” in the algorithm. These “gates” have the purpose of resolving the gradient problem, and they allow more precise control over the information that is kept in its memory [17]. Considering the highly dynamic behavior of the EEG signals, often the LSTM networks offered significantly better performance over different applications of EEG signals, such as emotion recognition, confusion detection and decision-making predictions [18,19,20]. A great success of LSTM neural networks for articulated speech recognition from EEG signals was presented in [21] for an automatic speech recognition (ASR) system. The researchers used MFCC as features and predicted the coefficients using different types of recognition systems: generative adversarial neural networks (GAN), Wasserstein generative adversarial neural network (WGAN) and LSTM Regression. The results showed an average of the root mean square (RMS) of 0.126 for the LSTM regression compared with 0.193 and 0.188 registered for the GAN and WGAN networks, respectively.

The most significant results from the state-of-the-art, regarding the imaginary speech recognition systems using surface EEG of the Kara One Database are presented in Table 1, along with the most relevant characteristics of the systems: pre-processing method, feature extractions and the classifier used.

**Table 1 sensors-22-04679-t001:** State-of-the-art EEG speech recognition of Kara One database phonemes and words.

Source	Task	Pre-Processing Method	Feature Extraction	Classification Method	Accuracy
[7]	Imagined speech: Vocal vs. Consonant (C/V) Presence of nasal (±Nazal) Presence of bilabial (±bilabial) Presence of /iy/ (±/iy/) Presence of /uw/ (±/uw/)	Eliminating ocular artifacts using Blind Source Separation (BSS) Band-pass filter 1–50 Hz	Features of window 10%/50% overlap: mean, median, standard deviation, variance, maximum, minimum, maximum ± minimum, sum, spectral entropy, energy, skewness and kurtosis	SVM-quad; Leave-one-out	C/V: 18.08% ±Nazal: 63.50% ±Bilabial: 56.64% ±/iy/: 59.60% ±/uw/: 79.16%
[9]	Imagined speech; Vocal vs. Consonant (C/V) Presence of nasal (±Nazal) Presence of bilabial ±bilabial) Presence of /iy/ (±/iy/) Presence of /uw/ (±/uw/)	Eliminating ocular artifacts using Blind Source Separation (BSS) Band-pass filter 1–50 Hz Subtraction of mean value from each channel	Cross-Covariance Matrix (CCV)	CNN + LTSM + Deep Autoencoder; Random shuffled data in train-validation-testing: 80-10-10; Cross-validation method	C/V: 85.23% ±Nazal: 73.45% ±Bilabial: 75.55% ±/iy/: 73.30% ±/uw/: 81.99%
[10]	Multi-class classification: /iy/, /uw/, /piy/, /tiy/, /diy/, /m/, /n/ + “gnaw”, “knew”, “pat”, “pot”	Band-pass filter 1–50 Hz Laplacian filter over each channel Window 500 ms + 250 overlap ICA for noise removal	Linear features: mean, absolute mean, standard deviation, sum, median, variance, max, absolute max, min, absolute min, max + min, max − min Non-linear features: Hurst exponent, Higuchi’s algorithm of fractal dimension, spectral power, spectral entropy, magnitude and phase MFCC coefficients	Decision tree; SVM; 5-fold Cross-validation; Patient specific	Multi-class: MFCC + decision tree: 19.69% Linear features + decision tree: 15.91% Non-linear features + decision tree: 14.67% MFCC + SVM: 20.80%
[11]	Multi-class classification: /iy/, /uw/, /piy/, /tiy/, /diy/, /m/, /n/ + “gnaw”, “knew”, “pat”, “pot”	Notch filter 60 Hz Band-pass filter 0.5–100 Hz Visual analysis of signals and eliminating noisy ones	MFCC coefficients	CNN Random shuffled data in train-validation-testing: 80-10-10;	Multi-class accuracy: 24.19%
[8]	Imagined speech and spoken EEG signals; Multi-class classification: /iy/, /uw/, /piy/, /tiy/, /diy/, /m/, /n/ + “gnaw”, “knew”, “pat”, “pot”	Band-pass filter 1–200 Hz, Subtraction of mean value from each channel	Imagined-EEG signals and phonemes and spoken EEG signals	NES-G model; Leave-One-Out	Multi-class accuracy: 41.5%
[12]	Imagined speech: Vocal vs. Consonant (C/V) Presence of nasal (±Nazal) Presence of bilabial (±bilabial) Presence of /iy/ (±/iy/) Presence of /uw/ (±/uw/) Multi-class classification: /iy/, /uw/, /piy/, /tiy/, /diy/, /m/, /n/ + “gnaw”, “knew”, “pat”, “pot”	Eliminating ocular artifacts using blind source separation (BSS) Band-pass filter 1–50 Hz Subtraction of mean value from each channel	Cross-covariance matrix (CCV)	Three hierarchical levels: 1. CNN + TCNN 2. DAE 3. Extreme gradient boost leave-one-out	C/V: 89.16% ±Nazal: 78.33% ±Bilabial: 81.67% ±/iy/: 87.20% ±/uw/: 85.00% Multi-class accuracy (without phonological features): 28.08% Multi-class accuracy (with phonological features): 53.34%
[This study]	Multi-class classification: /iy/, /uw/, /piy/, /tiy/, /diy/, /m/, /n/ + “gnaw”, “knew”, “pat”, “pot”	Notch filter 60 Hz Visual analysis of signals	Cross-covariance matrix in time-domain Cross-covariance matrix in frequency-domain Windows length: 0.25 s, 0.5 s and 1 s	CNN with different architectures 50%/50% of windows for training/testing	Best multi-class accuracy: 37.06%

This paper contains a study of EEG signals with the main purpose of recognizing seven phonemes and four words acquired during the development of the Kara One database. Our study was conducted on eight different subjects and is meant to be a subject’s shared system. By a subject’s shared system, we mean a system that can only be used by subjects in the database. However, it is not a subject-specific device that would require different training for each new subject but assumes that only a fine-tuning will be performed when adding a new subject.

This paper also aims to develop a study of two different features computed over different windows of a signal. We used as feature extraction the cross-covariance over the channels in time and frequency domains for data reduction and to encode the variability of the electrodes during the imaginary speech. This hypothesis is based on the fact that speaking is a complex mechanism, implying the connectivity of different areas of the brain during the entire process. We also studied the results obtained after applying a mean filter over the spectrum band with different window dimensions (3 and 5 samples).

Another study conducted in this paper was based on analyzing three different timeframes: 0.25, 0.5 and 1 s. Regarding this study, we aimed to determine the best analysis window dimension for EEG imaginary speech phoneme and word recognition. In a time series, the statistics of the entire signal is different from the statistics of smaller windows—a fact that can lead to a significant impact on the final results of the system.

In the second part of the study, we focused on different CNN architectures for feature classification in order to determine which one fits our data best.

## 3. Materials and Methods

### 3.1. Preparing Database

In this paper, we used the Kara One database described in [7]. The database contains signals acquired from 12 healthy subjects in 14 sessions during rest, speaking and thinking eleven different stimuli from which seven are phonemes (/iy/, /uw/, /piy/, /tiy/, /diy/, /m/, /n/) and four are words (“gnaw”, “knew”, “pat”, “pot”). Each prompt was presented 12 times, meaning a total of 132 recorded signals for each subject, except for the subjects MM05 and P02, with a total of 165 trials.

The signals were acquired following a given protocol in order to obtain repeatability in the database signals. The protocol started with a 5 s state of rest in which the subject needed to relax for the next stage. Afterward, the stimulus appeared on the prompt for 2 s, and the utterance of the prompt was heard by the subject. This was followed by a 5 s stage in which the subject was instructed to imagine speaking the prompt. Finally, the subject was also asked to speak the prompt aloud.

Our goal was to identify the imagined speech, so in this paper, we only used the signals corresponding to the 5 s state of imaginary speaking of the prompt. Next, we eliminated the first and last 0.5 s of the signal, considering that these intervals correspond to a transition state, obtaining a 4 s signal in the end.

The signals resulting from the database were visually analyzed by an expert. In the first step of visual data analysis, it was discovered that six of the fourteen sessions presented signals with high noises or unattached ground wires. This situation was also discussed by the developers of the database, Shunan Zhao and Frank Rudzicz, in their paper [7]. Considering that discovery, we discarded all signals from the six contaminated sessions. Afterward, the expert visually analyzed all signals corresponding to thinking indexes and eliminated from the study the ones with high noise contamination. After this process of data analysis, we finally obtained a database with 624 signals to work with during the study. All signals from the database were collected using the 10-20 system for electrode positioning. In this paper, we used 62 electrodes. The electrodes and their position in the 10-20 system used are detailed in [7]. Finally, the signals were filtered using a notch filter in order to remove the 60 Hz power line artifact and all multiples of 60 Hz smaller than the Nyquist frequency.

### 3.2. Feature Extraction

In the feature extraction stage, we aimed to analyze the performance of the system when using the time- versus frequency-domain feature extraction methods for silent speech recognition. Another comparison study conducted in this stage was based on computing the features using different timeframes: 0.25, 0.5 and 1 s without overlapping. During this study, we aimed to find the time window in which the signal is quasi-stationary, but also contains all the needed information regarding the utterance.

All signals were segmented using these timeframes, and 50% of the timeframes from each recording were randomly distributed in the training set and 50% in the testing set.

EEG data usually produce a high-dimension time series due to the multiple electrodes. To decrease the dimension of EEG data, usually a data compression stage is conducted based on feature selection in order to extract the essential information from the signals [10] or to reduce the number of channels based on their informational relevance in relation to the system goal [22]. A new approach to reducing the data dimension of the EEG signals was presented by Pramit Saha and Sidney Fels in their study [9], where they computed the cross-covariance between the channels in the time domain in order to encapsulate the variability of the electrodes. In this study, we also used this technique of feature extraction and expanded it in the frequency domain.

The cross-covariance between two channels (c1 and c2) was defined in this study as:(1)$$\mathrm{Cov}\left(X^{c1}\left(t\right),X^{c2}\left(t\right)\right)=E\left[\left[{\;X}^{c1}\left(t\right)-E(X^{c1}\left(t\right)\right]\left[{\;X}^{c2}\left(t\right)-E(X^{c2}\left(t\right)\right]\right],$$
where $X^{c1}\left(t\right)$ represents the EEG signal acquired for channel c1, $X^{c2}\left(t\right)$ is the EEG signal acquired for channel c2, and E[$X^{\mathrm{ch}}\left(t\right)$] represents the expected value (where ch corresponds to the specific channel c1 or c2) and is computed as:(2)$$E(X^{\mathrm{ch}}\left(t\right))=\frac{1}{W}\overset{W-1}{\underset{i=0}{\sum }}x_{i}^{\mathrm{ch}}$$

The W value of Equation (2) corresponds to the window dimension for which the features are computed.

The second method of feature extraction analyzed in this paper assumes the transformation of the time domain series of EEG signals into the frequency domain using the Fast Fourier Transform (FFT). The Fourier transform is a method used to decompose the signal into sinus and cosine waves.

The FFT of a channel was computed using the following:(3)$${\mathrm{FX}}^{\mathrm{ch}}\left(f\right)=\overset{n-1}{\underset{t=0}{\sum }}X_{t}^{\mathrm{ch}}e^{-\frac{j2\pi \mathrm{ft}}{n}}$$
where $X_{t}^{\mathrm{ch}}$ represents the EEG signal acquired for channel ch.

After computing the signals corresponding to the frequency-domain of desired channels using Equation (3), we computed the cross-covariance between the Fourier transform of the channels.

Figure 1 and Figure 2 present examples of a 2D feature matrix with a 62 × 62 dimension, corresponding to the time and frequency domain, respectively, for a 0.25 s window timeframe.

### 3.3. Classification

Convolutional neural networks (CNN) are powerful networks when applied to images. They have the power to understand the image content and to extract the deep information encoded in the input data. Nowadays, many systems are based on this type of neural network. CNN showed a great success in understanding biomedical images for classification, segmentation, detection and localization [23] for different types of input images, offered a great false prediction rate in seizure prediction systems based on EEG signals [24], and is widely used in BCI systems for imaginary motion recognition [25,26,27] and assisting in the diagnosis of Parkinson’s disease [28]. In the imaginary speech recognition domain, the CNN was a great resource for EEG signal classification [9,27].

The great success of CNN is due to the design of the hidden convolutional layers working as a decoder for the disguised essential information of the two-dimension matrix offered as input. It has the power to extract features and feed them to the dense layers designed to classify these computed features. The component of a CNN starts with an input layer that receives the given data. Then, it continues with the hidden layers corresponding to the convolutional layers in the first phase, which interprets the data received from the input. The output of the last convolutional layer is flattened and introduced into one or multiple dense layers having the purpose of learning the extracted features. Finally, the neural network contains an output layer, which usually has the role of classifying the data into the desired classes [29,30,31]. A general CNN block diagram is presented in Figure 3.

In our research, we tested different architectures of the CNN neural network with the purpose of finding the one offering the best performance with respect to the complexity, memory, and the running time. We started with a low complexity architecture with one convolutional layer and one dense layer, and we increased the complexity up to three convolutional layers and one dense layer, having a larger number of filters and neurons.

In the training phase, we used a learning rate of 0.0001, categorical cross-entropy as loss and Adam as optimizer. We divided the training set into 75% training and 25% validation and used k-fold cross-validation in order to obtain a more accurate performance result. Figure 4 presents an example of the architecture used in the classification stage, with two convolutional layers of 64 and 128 filters, respectively, and one dense layer with 64 neurons.

## 4. Results

During the development of the system, we aimed to study five different variables capable of influencing the performance of the imaginary speech recognition. Our study of system performance analysis included: (a) the influence of CNN hyperparameters; (b) modification of the network architecture; (c) the impact of the different activation functions that can be used in the CNN; (d) different features capable of encoding the speech hidden information by computing the covariance of the signals over the channels in time and frequency (B0) domain; (e) different window dimensions for the feature extraction method; (f) average filter of three (B3) and five (B5) dimension kernels over the computed spectrum of the data.

For further simplification of displaying the results of different architecture models, we used the abbreviation explained in Table 2. As an example, the architecture Conv2D (64, 128, 64)-Dense (64) corresponds to a CNN network with three convolutional layers, with 64, 128, and 64 filters, respectively, and one dense layer with the number of neurons in the layer equivalent to 64. For all architectures, after the dense layer was introduced, the output layer with 11 neurons corresponded to the 11 different classes.

The Kara One database does not show a significant class imbalance. The number of the samples from each class starts from a minimum of 83 (phoneme \m\) and reaches a maximum of 95 (word “pot”) out of a total of 993. The a priori probability rises from 0.083 for \m\ phoneme to 0.095 for the word “pot”.

### 4.1. Comparison of Activation Function: Tanh vs. Relu

The results obtained over the test set using different architectures of the CNN and different activation functions for the convolutional layers (hyperbolic tangent vs. rectified linear unit) using the covariance of the spectrum without an average filter (B0) computed over 0.5 s windows are detailed in Table 3.

### 4.2. Comparison of Features: Time vs. Frequency

Further in our study, we also compared the differences between the features computed over the signal in the time and frequency domains. The results obtained using different tested architectures are presented in Table 4. It is easy to observe a significant accuracy decrease when using time-domain cross-covariance versus frequency-domain features. The difference between the accuracy of the two feature extraction methods increases to approximately 16%, with the accuracy of frequency features reaching a maximum of 37% and the maximum accuracy of the time-domain features decreasing to 21%. These differences imply that information of speech is more easily decoded by the neural network in the frequency domain rather than in the time domain. The main advantage of the covariance in the frequency domain is given by the elimination of the possible delays of the stimulus propagation over the channels, starting from the source activation of the specific imaginary articulation of the phoneme.

A study of different architectures of the neural network shows (Figure 5) that a CNN with three convolutional layers with 64 and 128 filters and connected with a dense layer with 64 neurons works best for the frequency-domain features (the features that provided the best accuracy rate), obtaining a performance of 37% accuracy. When it comes to the time domain, the best results were obtained using less complex architectures, and the best performance of the system was recorded using only one convolutional layer with 64 filters and one dense layer with 64 neurons.

The mean confusion matrices for all k-folds for the time and frequency features are presented in Figure 6. In both images, we can see a distinction between phonemes and words. The system has a difficult time recognizing one phoneme against the other but makes a clearer distinction between them and the words. We can also observe that phoneme \diy\ is often confused with similar phonemes such as \iy\ and \piy\. It can also be seen that there is no significant imbalance in the recognition of any of the phonemes and words; however, the words have a higher accuracy rate of recognition.

### 4.3. Comparison of Time Window Length: 0.25, 0.5 and 1 s

After we concluded that the system works better with a rectified linear unit as an activation function for the convolutional layers in the frequency domain, we tested the network with different window lengths for the input data. The results are presented in Table 5.

The 4 s EEG signal containing imaginary speech includes multiple imaginations of the specific stimulus. It is hard to precisely determine the moment containing the desired signal in the whole four seconds of recording, which is why we chose to segment the signal over different window lengths and observe the system behavior. Table 5, as well as Figure 7, shows that the best analysis window is 0.25 s, reaching an accuracy of 37%. Looking at the 0.5 and 1 s window lengths, we can observe that the 0.5 s offered an accuracy close to the 0.25 s window, meaning that the signals are still easier to decode compared to the 1 s window in which the accuracy significantly dropped to 29%. The mean confusion matrices for the 0.5 s window and 1 s window are presented in Figure 8.

### 4.4. Comparison of Mean Filter Kernel: B0, B3 and B5

Another study conducted in this paper focused on applying different average filter lengths over the spectrum before computing the covariance matrix. We tested two different filter lengths: three samples and five samples. We will further refer to the spectrum without a mean filter as B0, the spectrum with an applied mean filter length of three samples as B3, and the spectrum with an applied mean filter length of five samples as B5. The obtained results can be seen in Table 6. The main motivation for this approach was developed on the assumption that the analysis of multiple values of the spectrum, as opposed to analyzing only the local values, can offer a better perspective of the frequency distribution regarding different classes. This assumption did not stand up because, as can be seen in Table 6 and in Figure 9, the better accuracy results were obtained using the unmodified spectrum. These results imply that every frequency is important for the phoneme and word recognition problem.

### 4.5. Performance Evaluation Metrics

For a better understanding of the recorded results and the system performance, we introduced the computed values for all extracted features: the balanced accuracy, kappa and recall [32]. The obtained values are presented in Table 7.

According to Table 7, the balanced accuracy and recall are not significantly different from the computed accuracies for the features, and only the kappa score dropped to a value of approximately 0.7 for all features.

### 4.6. Complexity and Memory Measurements

This paper aimed to develop a low-cost system working with limited resources. To achieve this goal, we tested different architectures of CNN networks for different types of features and windows. This research helped us to determine the best CNN architecture, features and window frame that can be implemented on a device with limited resources.

Given the application, the most significant resource consumer is the neural network. For the CNN neural network architecture, the complexity of the algorithm can be estimated as O(k × N × M × nF^L−1^ × nF ^L^), where k is the kernel matrix, N is the number of lines of the input matrix, M is the number of columns of the input matrix, nF^L−1^ is the number of filters from the anterior CNN layers and nF^L^ is the number of filters from the current layer. In our case, the input matrix has the same number of lines and columns (M = N = 62), and we can write the complexity as O(k × N^2^ × nF^L−1^ × nF ^L^). The details of the complexity, memory and time for the feature extraction stage and the best performance CNN architecture are presented in Table 8.

Using an AMD Ryzen 7 4800HS CPU with 16 GB memory RAM and 2.9 GHz clock frequency, we managed to obtain an average time per recognized input vector of 1.8 × 10^−3^ s starting from the feature extraction stage up to the decision making. The time was estimated (Table 8) using the characteristics of the best system in terms of performance, meaning computing the output for the 0.25 s window vector with the C64-128/D64 neural network architecture (Table 5).

A comparison of the methods in terms of execution time, as can be observed in Table 9, show that there are no significant differences between the execution of the different features; however, there is approximately an order of magnitude between the best performance architectures and the most complex one tested.

## 5. Discussion

This paper aims to compare different parameters of an intelligent imaginary speech recognition subject’s shared system to observe the performance variation when using different mechanisms of feature extraction and different architectures of CNN in the classification stage.

We used the Kara One database in our study, designed and conducted at the Toronto Rehabilitation Institute by Shunan Zhao and Frank Rudzicz [7], which contains signals acquired during speech and imaginary speech of seven phonemes and four words.

During the recognition process, we pre-processed the signals, and after the visual inspection, eliminated all data from subjects containing electrodes with bad connectivity and the signals with high noise. Furthermore, in the pre-processing stage, we applied a notch filter to remove the 60Hz power line artifact and all multiples of 60Hz smaller than the Nyquist frequency. It is worth mentioning that, in our study, we kept all high-frequency information.

### 5.1. Time vs. Frequency Features 

After the pre-processing stage, we went through a feature extraction stage where we focused on comparing the feature extraction based on cross-covariance over the channels in the time and frequency domains. The cross-covariance method is based on the fact that speech is a complex mechanism, requiring thinking of the speech stimulus, preparing the vocal tract for the actual vocalization and giving the signal to all components of the vocal tract involved in the actual speaking of the stimulus. For different stimuli, there are different positions and components involved in the process. This mechanism demands the activation of multiple areas of the brain that communicates in a very short time. The connections of different areas are best highlighted by the cross-covariance between the channels. The results presented in Table 4 show that there is a considerable difference between the results obtained using time-domain feature extractions versus frequency-domain feature extractions. When using frequency-domain features, the accuracy increases by approximately 16% to a value of 0.37 compared to 0.21 obtained when using features in the time-domain. This difference is given by the fact that the signal spectrum eliminates the delays of the stimulus propagation over the channels, starting from the activation focus of the specific imaginary articulation of the phoneme.

### 5.2. Time-Window Analysis

Another study conducted in this paper aims to compare different sizes of the analysis window in order to observe the signal statistics of different time gaps. During this study, we aimed to find the time window in which the signal is quasi-stationary but also contains all the needed information regarding the utterance. We compared three analysis window sizes: 0.25, 0.5 and 1 s. The obtained results can be seen in Table 5. Comparing the window dimensions, we observed that the best time window length was 0.25 s. The accuracy of the results is significantly higher when using 0.25 s, increasing to a value of 0.37, compared to 0.29 when using a 1 s window. The difference between the accuracy of the 0.25 s window and the 0.5 s window is 1%, which is not very significant. This means that for a 0.5 s window, the utterance of the phonemes and words are still captured by the frame.

Analyzing the results in Table 5 also shows that the maximum accuracy for all timeframe windows was obtained using a low-complex architecture for the CNN.

### 5.3. Mean Filter over the Spectrum Analysis

During our research for improvement, we also tried to average the spectrum of the signals with a filter of three and five samples. The main motivation for this approach was developed on the assumption that the analysis of multiple values of the spectrum, compared to analyzing only the local values, can offer a better perspective of the frequency distribution regarding different classes. The details of this research are presented in Table 6. As can be seen, applying an average filter over the spectrum did not increase the accuracy; on the contrary, the accuracy dropped by approximately 9% when using filters with three and five samples.

### 5.4. CNN Architectures Analysis

In our final study, we tested different architectures for the CNN network to observe the system performance and shape the way for the future development of similar systems. We concluded that when it comes to the frequency-domain features (the features that provided the best accuracy rate), the best architecture is two convolutional layers with 64 and 128 filters connected to a dense layer with 64 neurons. More complex architectures do not improve the performance of the system, and on the contrary, the performance decreases.

## 6. Conclusions

This paper analyses the EEG signals for imaginary speech recognition of seven phonemes and four words. To accomplish our purpose, we developed an intelligent subject’s shared system using a processing chain applied to the Kara One database [7]. The first stage in the analysis chain started with pre-processing the input signals in order to obtain better quality data. Further in the feature extraction stage, we compared the results obtained after computing the cross-covariance over the channels in the time and frequency domains. During our research, we also studied different time window lengths: 0.25, 0.5 and 1 s to find the time window in which the signal is quasi-stationary but also contains all the information needed regarding the utterance. We also studied the system behavior when applying a mean filter with kernel sizes of three and five samples assuming that the analysis of multiple values of the spectrum, compared to analyzing only the local values, can offer a better perspective of the frequency distribution regarding different classes. Finally, in the classification stage, we tested multiple architectures of the CNN neural network to determine the best performance of the system.

The best results were obtained using the cross-covariance over channels in the frequency domain using a 0.25 s window length. The best performance of the system was recorded when using a CNN with two convolutional layers and 64 and 128 filters, connected to a dense layer with 64 neurons. With these system characteristics, we achieved an accuracy of 37%, a significant improvement compared to using the Mel-Cepstral Coefficients for feature extraction, where the best accuracy recorded was 20.80% when using an SVM as the classifier [10] and 24.19% when using a CNN as the classifier [11]. During our study, we also showed that cross-covariance in the frequency domain offers a better understanding of the imaginary speech, reporting a better accuracy in comparison to the study made by Pramit Saha, Muhammad Abdul-Mageed and Sidney Fels in [12] where, using the cross-covariance in time and hierarchical deep learning (without phonological features), the best reported accuracy was 28%. However, when using phonological features, the accuracy increased to 54%, but this compromised the complexity and the memory of the system and is more difficult to implement in a low-complexity portable device.

The main limitation of our proposed system includes the acquisition of new data for each new subject before being able to wear the system. The collected data must be included in the database for which a fine-tuning of the network training must be applied. However, this limitation can be overcome in time by enriching the database with new examples.

In this study, we proposed a feature extraction method based on cross-covariance in the frequency domain that offered a significant improvement for the system performance compared to features computed in the time domain. We are confident that these features can be further exploited to obtain even more precise systems for imaginary speech recognition.

In this paper, we achieved our goal of highlighting the importance of using frequency in the feature extraction stage in contrast to the time domain. The advantage of using the frequency domain is given by the elimination of the delays caused by the propagation of the stimulus from one channel to another during the imaginary articulation of the speech. We also showed that a quicker analysis of the signal offers a better understanding of the thinking speech.

Finally, we can say that the proposed system qualifies as a portable, low-cost system using limited resources for decision making. The running time for the best performance CNN architecture was 1.8 ms tested on an AMD Ryzen 7 4800HS CPU.

## Figures and Tables

**Figure 1 sensors-22-04679-f001:**
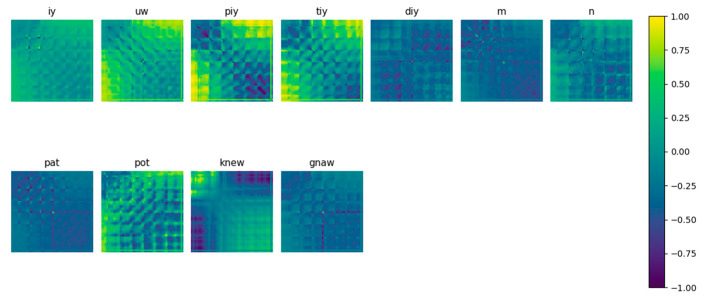
Example of a 2D feature matrix computed in the time domain for 0.25 s time window.

**Figure 2 sensors-22-04679-f002:**
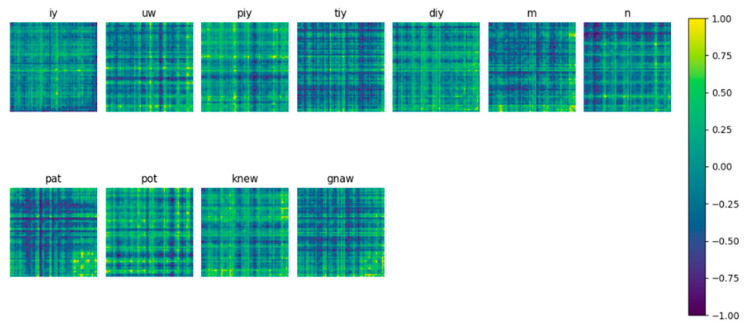
Example of a 2D feature matrix computed in the frequency domain for a 0.25 s time window.

**Figure 3 sensors-22-04679-f003:**
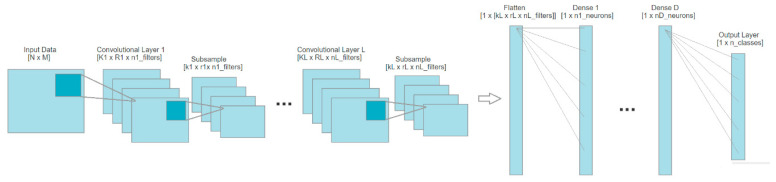
Block diagram of general convolutional neural networks.

**Figure 4 sensors-22-04679-f004:**
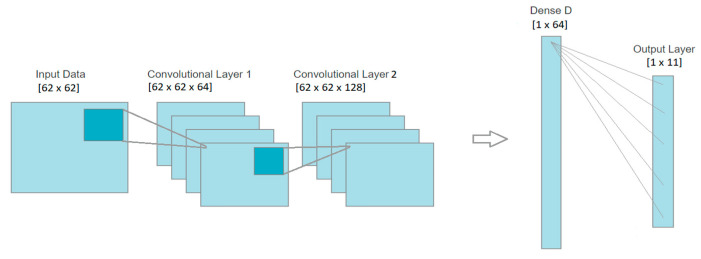
Block diagram of the CNN architecture used in the classification stage.

**Figure 5 sensors-22-04679-f005:**
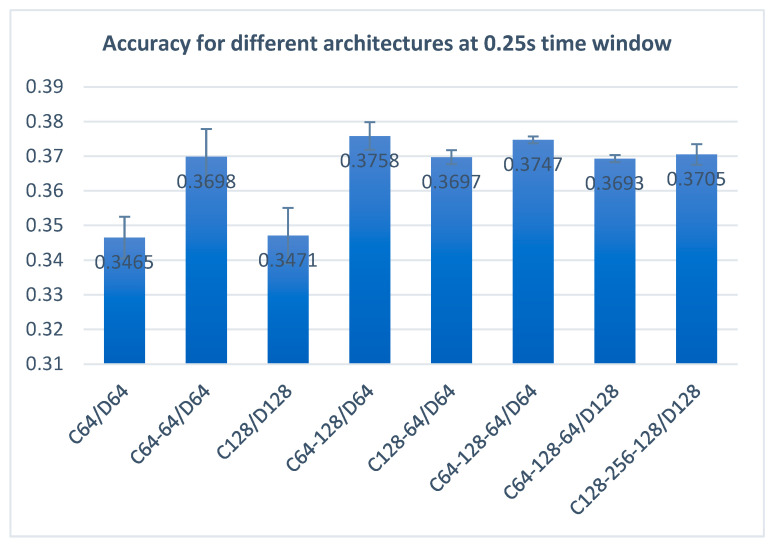
Accuracy for different architectures with a 0.25 s time window.

**Figure 6 sensors-22-04679-f006:**
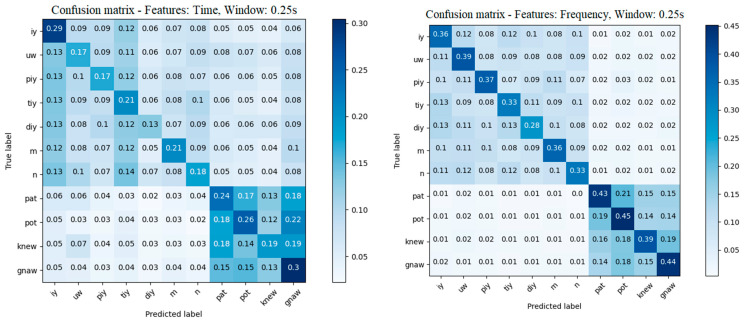
Mean of k-fold confusion matrix in the time and frequency domains.

**Figure 7 sensors-22-04679-f007:**
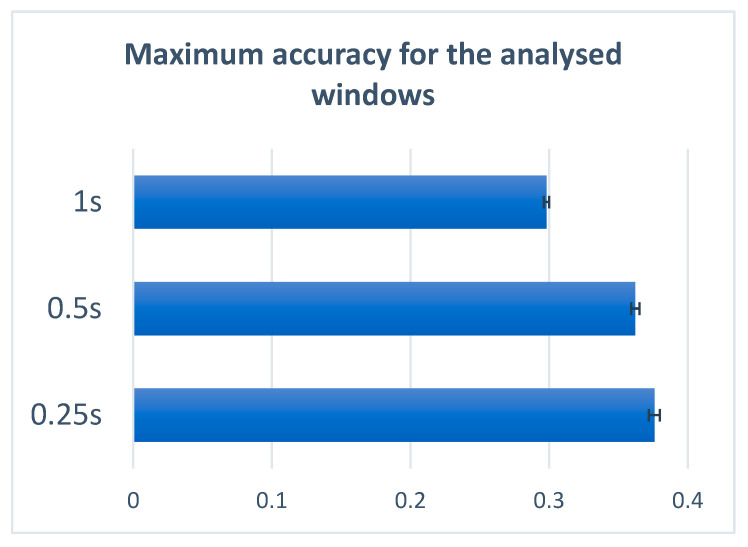
Maximum accuracy for the analyzed windows.

**Figure 8 sensors-22-04679-f008:**
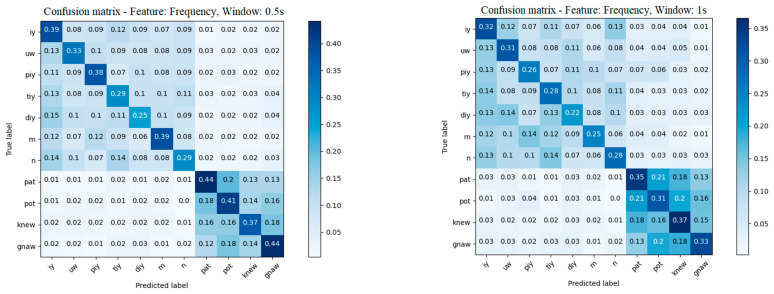
Mean of k-fold confusion matrix in the frequency domain for 0.5 s and 1 s windows.

**Figure 9 sensors-22-04679-f009:**
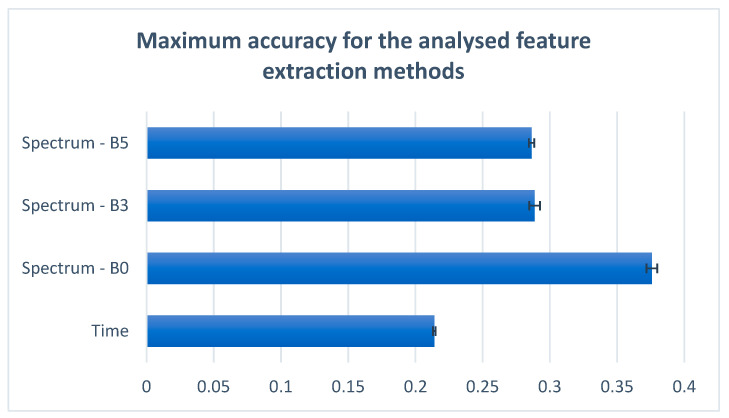
Maximum accuracy for the feature extraction methods analyzed.

**Table 2 sensors-22-04679-t002:** Convolutional Neural Network architecture abbreviations.

Architecture	Abbreviation
Conv2D (64)-Dense (64)	C64/D64
Conv2D (64, 64)-Dense (64)	C64-64/D64
Conv2D (128)-Dense (128)	C128/D128
Conv2D (64, 128)-Dense (64)	C64-128/D64
Conv2D (128, 64)-Dense (64)	C28-64/D64
Conv2D (64, 128, 64)-Dense (64)	C64-128-64/D64
Conv2D (64, 128, 64)-Dense (128)	C64-128-64/D128
Conv2D (128, 256, 128)-Dense (128)	C128-256-128/D128
Conv2D (512, 256, 128)-Dense (128)	C512-256-128/D128

**Table 3 sensors-22-04679-t003:** Results obtained using different CNN architectures for the covariance of spectrum features computed over a 0.5 s window comparing the hyperbolic tangent activation function of convolutional layers with the rectified linear unit.

Data characteristics: Features: Covariance of Spectrum Window: 0.5 s Bands: B0	Convolution Layer—Tanh Dense Layer—Tanh Output Layer—Softmax		Convolution Layer—Relu Dense Layer—Tanh Output Layer—Softmax	
	Loss	Accuracy	Loss	Accuracy
C64/D64	1.9344 ± 0.022	0.2974 ± 0.003	1.7929 ± 0.015	0.3465 ± 0.006
C64-64/D64	1.9549 ± 0.020	0.2939 ± 0.007	1.9581 ± 0.135	0.3698 ± 0.008
C128/D128	1.9146 ± 0.059	**0.3169 ± 0.004**	1.7818 ± 0.028	0.3471 ± 0.008
C64-128/D64	1.9459 ± 0.020	0.2932 ± 0.004	1.9514 ± 0.078	**0.3758 ± 0.004**
C128-64/D64	1.9374 ± 0.033	0.2954 ± 0.003	2.0107 ± 0.063	0.3697 ± 0.002
C64-128-64/D64	1.9634 ± 0.052	0.2901 ± 0.008	2.1169 ± 0.034	0.3747 ± 0.001
C64-128-64/D128	1.9882 ± 0.105	0.3035 ± 0.006	2.3308 ± 0.042	0.3693 ± 0.001
C128-256-128/D128	2.0120 ± 0.105	0.2989 ± 0.011	2.4393 ± 0.107	0.3705 ± 0.003

**Table 4 sensors-22-04679-t004:** The results obtained after computing the different feature extractions: in the time domain and frequency domain over windows of 0.25 s.

Data Characteristics: Activation Functions: Relu-Tanh-Softmax Window: 0.25 s	Time		Frequency	
	Loss	Accuracy	Loss	Accuracy
C64/D64	2.3402 ± 0.027	**0.2140 ±** **0.001**	1.7929 ± 0.015	0.3465 ± 0.006
C64-64/D64	2.4682 ± 0.075	0.2128 ± 0.003	1.9581 ± 0.135	0.3698 ± 0.008
C128/D128	2.5216 ± 0.155	0.2115 ± 0.002	1.7818 ± 0.028	0.3471 ± 0.008
C64-128/D64	2.3019 ± 0.031	0.2051 ± 0.001	1.9514 ± 0.078	**0.3758 ±** **0.004**
C128-64/D64	2.5804 ± 0.324	0.2071 ± 0.010	2.0107 ± 0.063	0.3697 ± 0.002
C64-128-64/D64	2.5197 ± 0.184	0.2038 ± 0.004	2.1169 ± 0.034	0.3747 ± 0.001
C64-128-64/D128	3.2871 ± 0.402	0.2039 ± 0.002	2.3308 ± 0.042	0.3693 ± 0.001
C128-256-128/D128	3.0037 ± 0.571	0.1981 ± 0.001	2.4393 ± 0.107	0.3705 ± 0.003

**Table 5 sensors-22-04679-t005:** Comparison of different window length (0.25, 0.5, 1 s) results for the covariance of spectrum features without an average filter (B0).

Data Characteristics: Features: Covariance of Spectrum Activation Functions: Relu-Tanh-Softmax Bands: B0	0.25 s		0.5		1 s	
	Loss	Accuracy	Loss	Accuracy	Loss	Accuracy
C64/D64	1.7929 ± 0.015	0.3465 ± 0.006	1.9221 ± 0.023	0.3243 ± 0.005	2.0960 ± 0.033	0.2808 ± 0.011
C64-64/D64	1.9581 ± 0.135	0.3698 ± 0.008	1.9642 ± 0.019	0.3514 ± 0.004	2.2721 ± 0.087	0.2939 ± 0.007
C128/D128	1.7818 ± 0.028	0.3471 ± 0.008	1.8257 ± 0.017	0.3400 ± 0.004	2.0517 ± 0.025	0.2857 ± 0.004
C64-128/D64	1.9514 ± 0.078	**0.3758 ± 0.004**	1.9957 ± 0.065	0.3588 ± 0.003	2.1825 ± 0.087	**0.2980 ± 0.002**
C128-64/D64	2.0107 ± 0.063	0.3697 ± 0.002	1.9903 ± 0.053	**0.3620 ± 0.003**	2.3020 ± 0.126	0.2964 ± 0.006
C64-128-64/D64	2.1169 ± 0.034	0.3747 ± 0.001	2.1737 ± 0.080	0.3566 ± 0.002	2.3971 ± 0.091	0.2922 ± 0.003
C64-128-64/D128	2.3308 ± 0.042	0.3693 ± 0.001	2.2874 ± 0.212	0.3457 ± 0.007	2.6291 ± 0.135	0.2938 ± 0.013
C128-256-128/D128	2.4393 ± 0.107	0.3705 ± 0.003	2.4186 ± 0.142	0.3504 ± 0.005	2.6111 ± 0.325	0.2925 ± 0.006
C512-256-128/D128	2.5051 ± 0.086	0.3680 ± 0.005	2.1755 ± 0.101	0.3136 ± 0.007	2.6600 ± 0.2066	0.2871 ± 0.008

**Table 6 sensors-22-04679-t006:** Comparison between the results obtained after applying different kernels for the average filter of the spectrum. The analysis window is 0.5 s.

Data Characteristics: Features: Covariance of Spectrum Activation Functions: Relu-Tanh-Softmax Window: 0.25 s	B0		B3		B5	
	Loss	Accuracy	Loss	Accuracy	Loss	Accuracy
C64/D64	1.7929 ± 0.015	0.3465 ± 0.006	2.0423 ± 0.011	0.2864 ± 0.003	2.0479 ± 0.023	0.2809 ± 0.003
C64-64/D64	1.9581 ± 0.135	0.3698 ± 0.008	2.4141 ± 0.185	0.2837 ± 0.007	2.3275 ± 0.023	0.2841 ± 0.003
C128/D128	1.7818 ± 0.028	0.3471 ± 0.008	2.0203 ± 0.041	**0.2886 ± ** **0.004**	1.9876 ± 0.025	0.2825 ± 0.003
C64-128/D64	1.9514 ± 0.078	**0.3758 ± 0.004**	2.2003 ± 0.121	0.2838 ± 0.003	2.3150 ± 0.119	**0.2863 ± 0.002**
C128-64/D64	2.0107 ± 0.063	0.3697 ± 0.002	2.2907 ± 0.150	0.2771 ± 0.005	2.3421 ± 0.215	0.2804 ± 0.005
C64-128-64/D64	2.1169 ± 0.034	0.3747 ± 0.001	2.5577 ± 0.244	0.2755 ± 0.003	2.5767 ± 0.171	0.2786 ± 0.007
C64-128-64/D128	2.3308 ± 0.042	0.3693 ± 0.001	2.4874 ± 0.226	0.2753 ± 0.001	2.5579 ± 0.3141	0.2707 ± 0.007
C128-256-128/D128	2.4393 ± 0.107	0.3705 ± 0.003	2.9467 ± 0.197	0.2758 ± 0.006	2.8533 ± 0.259	0.2707 ± 0.006

**Table 7 sensors-22-04679-t007:** The obtained result for all features of balanced accuracy, kappa and recall.

	Time— Window Length 0.25 s	Frequency—Window Length 0.25 s	Frequency—Window Length 0.5 s	Frequency—Window Length 1 s
Balanced accuracy	0.2131 ± 0.001	0.3749 ± 0.004	0.3615 ± 0.003	0.2980 ± 0.001
Kappa	0.1349 ± 0.001	0.3132 ± 0.004	0.2980 ± 0.004	0.2278 ± 0.002
Recall	0.2140 ± 0.001	0.3750 ± 0.004	0.3620 ± 0.003	0.2980 ± 0.002

**Table 8 sensors-22-04679-t008:** Detailed complexity, memory and time computation for the system with the best results.

System Stages		Complexity	Memory	Time (s)
Feature Extraction	FFT	O(NxMlogM)	968 KB	3.12 × 10^−4^
	COV	O(N^2^)	88 KB	4.84 × 10^−4^
CNN	Conv2D-64	O(k × N^2^ × 64)	976 KB	1.7 × 10^−3^
	Conv2D-128	O(k × N^2^ × 64 × 128)	2.81 MB	
	Dense-64	O(k × N^2^ × 128 × 64)	7.8 MB	
	Dense-11	O(64 × 11)	748 B	

**Table 9 sensors-22-04679-t009:** Execution time for all tested architectures and features.

Architecture	Features			
	Time— Window Length 0.25 s	Frequency— Window Length 0.25 s	Frequency— Window Length 0.5 s	Frequency—Window Length 1 s
C64/D64	5.4 × 10^−4^	6.4 × 10^−4^	6.3 × 10^−4^	7.3 × 10^−4^
C64-64/D64	1.1 × 10^−3^	1.2 × 10^−3^	1.2 × 10^−3^	1.3 × 10^−3^
C128/D128	1.5 × 10^−3^	1.8 × 10^−3^	1.6 × 10^−3^	1.7 x10^−3^
C64-128/D64	1.7 × 10^−3^	1.8 × 10^−3^	1.9 × 10^−3^	1.9 × 10^−3^
C128-64/D64	2 × 10^−3^	2.1 × 10^−3^	2.2 × 10^−3^	2.3 × 10^−3^
C64-128-64/D64	2.7 × 10^−3^	2.8 × 10^−3^	2.9 × 10^−3^	2.9 × 10^−3^
C64-128-64/D128	2.8 × 10^−3^	2.9 × 10^−3^	3 × 10^−3^	3.1 × 10^−3^
C128-256-128/D128	9.2 × 10^−3^	10^−2^	10^−2^	8.9 × 10^−3^

## Data Availability

http://www.cs.toronto.edu/~complingweb/data/karaOne/karaOne.html. (accessed on 16 June 2022).

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
