# Peer review of "CNN Architectures and Feature Extraction Methods for EEG Imaginary Speech Recognition"

_sensors, 2022, doi:10.3390/s22134679_

Round 1

Reviewer 1 Report

The topic is interesting. In the paper, the authors analyzed the EEG signals for imaginary speech recognition of seven phonemes and four words. They developed an intelligent subject-independent system using a processing chain applied to a database. The obtained results are encouraging. However, the following aspects should be considered to improve the structure and quality of the paper:

  1. In the abstract part, the novelty of the proposed approach should be described. The authors only described that “This paper aims to study different parameters of an intelligent imaginary speech recognition system to obtain the best performance according to the developed method” and “Another study included in the paper is based on comparison of different architectures of Convolutional Neural Network (CNN).” Of course, the reviewer can understand the aim and features of the proposed technique.
  2. The main contributions are unclear. Please introduce a paragraph where these are to be highlighted clearly.
  3. The authors should introduce the additional comments regarding Figures 3 – 8. 
  4. The effectiveness of this paper is not clear. Through experiments, the authors must justify the effectiveness of the proposed method by comparing it with the other conventional methods. Several methods are discussed in the research survey. Please show comparison data with these methods.
  5. The limits of the proposed approach should be better presented.

Reviewer 2 Report

The authors have performed hyper parameter tuning of a CNN model to perform speech decoding on a publicly available dataset. There is no novelty here also the performance achieved is not state of the art. The  idea for the feature has also proposed before. This doesn't inform anything new to the speech decoding community. Further, I suspect serious technical error in the feature engineering (heat maps are not scaled to min/max) and thus these methods are not appropriate. I will advise the authors to add more content to the paper and resubmit once they achieve state of the art performance. Some suggestions:

Introduction only focuses on EEG. There are several MEG and ECoG studies that have showed good performance in speech decoding. I would advise the author to talk about those in the literature survey.

Table 2 and figure 1 is not necessary to include in the manuscript. The data hey have used has that information so a simple citation would suffice.

Figure 2 doesn't say anything just a signal representation and not necessary to include in the manuscript.

Explaining the theory of Fourier transform is trivial not necessary to include in the manuscript.

Showing features in Figure 3-8 for different windows is just too much information. Probably select one time window and show both time and frequency representations. and definitely include colorer. Currently the features are not in the same scale or else you should see 1 in the diagonal of every heat map. if the authors have used these features to train the model the technical aspect of the manuscript is completely wrong. All of the heat maps has to be scaled in [-1 1] for each time window, trial, and domain (time/frequency) for this analysis to make sense.

Explaining the theory of CNN is not necessary. Architectural details are the only things that should be mentioned for reproducibility.

Table 4,5,6,7 and figure 8 just shows your hyper parameter tuning results which is again trivial. any deep learning training must follow tuning on the validation data but you have applied different architectures on the test data. This is not technically correct. You need to tune the hyper parameters on the validation data and use the best architecture to test your data. These tuning results are not interesting to publish but should go to the supplementary files.

Overall there is no new content except the frequency based cross covariance feature but that's not the best feature if it can't beat the state of the art performance.

Please consider these comments and try to rectify your paper. Reproducibility is always encouraged and if that's your gol try to use the publicly available codes for this data set, reproduce the results and compare your results. it is often advisable to use more than one publicly available data set to show that your feature and methods actually works.

Reviewer 3 Report

The main objective of the paper is a study for testing different configurations of convolutional neural networks (CNN) applied to electroencephalographic (EEG) imaginary speech recognition. The possible contribution of the paper is focused on experimental results since the methods employed are well known. The paper might be interesting from a practical standpoint as it contains extensive experiments. In general, explanations are comprehensive and figures are informative; however, there is still room for improvement in this regard. In addition, discussion on competitive methods to process EEG signals as well as evaluation of results should be improved. In summary, I consider the contents of the paper are potentially publishable, but the following issues should be addressed in a revised version of the paper.

-  Literal presentation has room for improvement. Theoretical information of well-known methods such as CNN and Fourier transform should be summarized or eliminated. For instance, several equations could be removed or included in the text.

- Performance evaluation of the proposed method should be improved by estimating the variability and statistical significance of the results. Thus, the mean and standard deviation of the Montecarlo experiments (or cross validation experiments) should be estimated and discussed and a statistical significance test of those variables should be applied.

- Long short-term memory (LSTM) recurrent neural networks were briefly commented in the paper. However, non-Gaussian mixtures-based methods and LSTM have demonstrated to be adequate and competitive to deal with EEG signals in problems similar to the proposed one. Non-stationary and non-linear changing of brain dynamics and classification with structured results were considered. Please discuss theoretically and/or practically on those methods for comparisons. I suggest the following reference: https://doi.org/10.1016/j.patcog.2019.04.022 .

- The following items should be considered to improve performance evaluation of the method: (i) others indices should be implemented such as balanced accuracy, kappa, and recall. (ii) Please include the a priori probability of the classes and discuss on possible imbalance of the data. (iii) The mean of the confusion matrices of the classification should be estimated, displayed, and discussed. (iv) A computational burden analysis of the experiments should be added including execution times for each of the implemented methods.

- Please improve readability of axis titles of figures 1-8.

Reviewer 4 Report

The goal of this work is to investigate various characteristics of an intelligent imaginary voice recognition system in order to achieve the best results using the created method. Signals including records for seven phonemes and four words were collected. This paper proposes comparison of two feature extraction algorithms based on obtaining the cross-covariance matrix in the time and frequency domain.  comparison of alternative CNN architectures is presented to track the system's evolution. The overall merit of the work is satisfactory and can be considered for publication however minor some concerns/revisions are required address. 1. Separate the introduction and literature review. 2. clearly specify your contributions in introduction section. 3. you claimed the study is for patient independent generalized system. How? 4. why conclusion of your work including final results are provided in introduction section. Here, write your contributions in few lines. 5. describe notch filter. 6. Figure 3-8, Is ti feature map or feature matrix? 7.  Figure 9. Block Diagram of general Convolutional Neural Networks. Is it general block diagram or your architetcure? provide your CNN architetcure. 8.  Table 3. The fixed parameters of the CNN. How can you fix parameter in CNN. you need to optimize the parameter from parameter space.  9. Compare results with some existing work.

Round 2

Reviewer 1 Report

The authors have tried and succeed to respond satisfactorily to each issue raised by the reviewer. Thus, they performed changes in the manuscript, and new explanations and elaborations of details have been brought. 

Author Response

Thank you very much for the time allocated to review the paper. We are glad we managed to approach all the raised issue and meet al the requirements at a satisfactory level. We also think that the comments made increased the quality of the paper and we are very grateful for helping us improving it.

Reviewer 3 Report

The quality of the paper has been significantly improved. All my concerns have been adequately addressed in the revised version including: improvement of the literal presentation of the paper; improvement of the performance evaluation and discussions on the results. Therefore, I consider the contents of the paper should be ready for publication.

Author Response

We are grateful for your review and for the comments. They helped us improve the quality of the paper. We are glad that we managed to address every comment at a satisfactory level, and we also thank you for the time allocated to review our paper.